# Low-Frequency Electrical Stimulation of the Auricular Branch of the Vagus Nerve in Patients with ST-Elevation Myocardial Infarction: A Randomized Clinical Trial

**DOI:** 10.3390/jcm14061866

**Published:** 2025-03-10

**Authors:** Sofia Kruchinova, Milana Gendugova, Alim Namitokov, Maria Sokolskaya, Irina Gilevich, Zoya Tatarintseva, Maria Karibova, Vasiliy Danilov, Nikita Simakin, Elena Shvartz, Elena Kosmacheva, Vladimir Shvartz

**Affiliations:** 1Scientific Research Institute of Regional Clinical Hospital #1 Ochapovsky, 350086 Krasnodar, Russianamitokov.alim@gmail.com (A.N.); giliv@list.ru (I.G.);; 2Department of Therapy #1, Kuban State Medical University, 350063 Krasnodar, Russia; gendugova@mail.ru (M.G.);; 3Bakoulev Scientific Center for Cardiovascular Surgery, 121552 Moscow, Russia; 4Autonomous Non-Profit Organization Sports School “Become a Champion”, 350063 Krasnodar, Russia; 5Cardiology Department, Novorossiysk City Hospital, 353915 Novorossiysk, Russia; 6National Medical Research Center for Therapy and Preventive Medicine, 101990 Moscow, Russia; shvartz.en@ya.ru

**Keywords:** myocardial infarction, reperfusion injury, vagus nerve stimulation, ST-elevation myocardial infarction, hospital mortality, cardiogenic shock, atrial fibrillation, stroke, transient ischemic attack, transcutaneous vagus nerve stimulation

## Abstract

**Background**: Despite the vast evidence of the beneficial effect of vagus nerve stimulation on the course of myocardial infarction confirmed in studies using animal models, the introduction of this method into actual clinical practice remains uncommon. **Objective**: The objective of our study was to evaluate the effect of transcutaneous vagus nerve stimulation (tVNS) on in-hospital and long-term outcomes for patients with ST-elevation myocardial infarction. **Materials and Methods**: A blind, randomized, placebo-controlled clinical trial was conducted. The participants were randomly split into two groups. The Active tVNS group was subjected to stimulation of the tragus containing the auricular branch of the vagus nerve. The Sham tVNS group underwent stimulation of the lobule. Stimulation was performed immediately on admission before the start of the percutaneous coronary intervention (PCI). Then, tVNS continued throughout the entire PCI procedure and 30 min after its completion. The primary endpoints were hospital mortality and 12-month mortality. The secondary endpoints were in-hospital and remote non-lethal cardiovascular events. The combined endpoint consisted of major adverse cardiovascular events (MACEs)—recurrent myocardial infarction, stroke/TIA, and overall mortality. **Results**: A total of 110 patients were randomized into the Active tVNS group (n = 55) and the Sham tVNS group (n = 55). The incidences of hospital mortality, cardiogenic shock, and AV block 3 were statistically less common in the Active tVNS group than in the Sham tVNS group (*p* = 0.024*, *p* = 0.044*, and *p* = 0.013*, respectively). In the long-term period, no statistical differences were found in the studied outcomes obtained following the construction of Kaplan–Meyer survival curves. When comparing groups by total mortality, taking into account hospital mortality, we observed a tendency for the survival curves to diverge (Logrank test, *p* = 0.066). Statistical significance was revealed by the composite endpoint, taking into account hospital events (Logrank test, *p* = 0.0016*). **Conclusions**: tVNS significantly reduced hospital mortality (*p* = 0.024*), the level of markers of myocardial damage, and the frequency of severe cardiac arrhythmias in patients with acute myocardial infarction. In the long term, the prognostic value of tVNS was revealed by the composite endpoint major adverse cardiovascular events. Further studies with an expanded sample are needed for a more detailed verification of the data obtained to confirm the effectiveness of tVNS and allow an in-depth analysis of the safety and feasibility of its use in routine clinical practice. This clinical trial is registered with ClinicalTrials database under a unique identifier: NCT05992259.

## 1. Introduction

Time is crucial in the treatment of acute myocardial ischemia, which highlights the importance of rapid diagnosis and timely revascularization. Prompt restoration of coronary blood flow minimizes ischemic damage and reduces the risk of complications of a myocardial infarction (MI) [1,2]. The key to success is the need to reduce the time from the onset of the ischemic process to blood supply restoration, which helps avoid the death of cardiomyocytes and prevent irreversible changes in the myocardium [3]. However, even in cases where coronary blood flow is restored within the golden hour, cascade processes of ischemia-related damage may take place: as a result of the cessation of blood supply to the myocardium, metabolic products accumulate, and free radicals form, thereby causing damage to cardiomyocytes [4].

Reperfusion injury is another important issue that arises when blood flow is restored after a long period of ischemia. It involves an acute inflammatory response accompanied by aggravated oxidative stress, which exacerbates existing damage. This condition requires additional therapeutic approaches aimed at neutralizing inflammatory processes and correcting metabolic activity in the myocardium [5,6,7]. From a pathophysiological standpoint, there is a complex relationship between ischemia and reperfusion injury, each of which affects the other, leading to further disruption of the myocardial structure and function.

Research on the cholinergic anti-inflammatory pathway provides an important contribution to understanding the interactions between the nervous system and the immune response, especially in the context of the pathophysiology of cardiovascular diseases [8]. Studies carried out in the 2000s formed the basis for a novel direction in investigating the role of the vagus nerve in the modulation of inflammatory processes [9,10,11]. These studies demonstrated that cholinergic neurons are capable of inhibiting the release of proinflammatory cytokines, an action achieved by activating the cholinergic alpha-7 nicotinic receptor on macrophages and other immune system cells [12,13]. This discovery was beneficial for the development of new treatment strategies for diseases associated with inflammation based on the modulation of vagus nerve activity. In particular, some studies employing animal models confirmed that vagus nerve stimulation (VNS) significantly reduces the size of MIs resulting from ischemia/reperfusion [14,15].

An important non-inflammatory effect of VNS is a reduction in sympathetic activity in the regulation of the cardiovascular system, which leads to a decrease in heart rate, a reduced risk of arrhythmia, and improved coronary blood flow. Parasympathetic dysfunction is a key factor contributing to adverse myocardial remodeling, the progression of heart failure, and the development of ventricular arrhythmias after MI, while VNS restores vagosympathetic balance.

Despite the vast evidence of the beneficial effect of VNS on the course of MI confirmed in studies using animal models, the introduction of this method into actual clinical practice remains uncommon [16]. In the only randomized clinical trial using humans [17], applying transcutaneous VNS (tVNS) to treat ST-elevation myocardial infarction (STEMI) reduced the frequency of ventricular arrhythmias, along with the concentrations of myocardial damage markers and inflammatory markers in the first few days after the development of MI.

Given such favorable outcomes, it is unclear why more similar and scaled-up studies have not been conducted yet. Therefore, additional clinical studies on the safety and efficacy of tVNS in acute MI are direly needed. The objective of our study was to evaluate the effect of tVNS on the in-hospital and long-term outcomes (at 12 months) of patients with STEMI.

## 2. Materials and Method

### 2.1. Study Design

We conducted a single-blind, randomized, placebo-controlled clinical trial, namely, Auricular Vagus Stimulation and STEMI, registered at http://clinicaltrials.gov (with the latest access being on 10 February 2025) with the following unique identifier: NCT05992259. The study protocol complied with the 1975 Declaration of Helsinki and the Ethical Guidelines for Epidemiological Research developed by the Russian Government and was approved by the local Ethics Committee of the State Budgetary Healthcare Institution, Ochapovsky Regional Clinical Hospital, Krasnodar, Russia.

Written informed consent was obtained from each patient prior to randomization procedure. This study was designed parsimoniously and pragmatically in order to maximize the likelihood of its further practical application. The data were collected by the authors at Ochapovsky Regional Clinical Hospital and then reviewed and processed by independent analysts at Bakulev Scientific Center for Cardiovascular Surgery and National Medical Research Center for Therapy and Preventive Medicine, Moscow, Russia.

### 2.2. Inclusion Criteria

Patients aged 40 to 90 years (adults and seniors);Patients afflicted with primary STEMI;Patients who had been treated in the first 12 h from the onset of pain;Patients who had undergone a primary percutaneous coronary intervention (PCI);Patients who had signed a voluntary informed consent form to participate in this study.

### 2.3. Non-Inclusion Criteria

A history of MI;Acute heart failure (grades III–IV according to NYHA);Bradyarrhythmia;Atrial fibrillation/flutter at the time of physical examination;Thrombolytic therapy at the prehospital stage;PCI/coronary artery bypass grafting (CABG) in the anamnesis;Participation in another clinical trial as a patient.

### 2.4. Exclusion Criteria

PCI cancellation or a failed PCI;Emergency change of PCI to CABG.

### 2.5. Randomization

Participants were randomly divided into two groups. Active tVNS group was subjected to stimulation of the tragus containing the auricular branch of the vagus nerve. Sham tVNS group underwent stimulation of the lobule, which does not contain the auricular branch of the vagus nerve. Random assignment to treatment groups was performed using an automated system in a mobile application on the smartphone of the principal investigator.

This was a single-blind study. It was difficult to ensure double-blindness due to technical limitations. Patients were urgently admitted to the hospital at any time of the day. The on-call doctor in the emergency cardiology department, who participated in the study, decided whether the patients met all the inclusion and exclusion criteria. Therefore, after randomization, the same doctor installed the tVNS device and accompanied the patients throughout the stimulation procedure. Of course, he knew which intervention group each patient belonged to. However, this did not affect the assessment of the results, as the patients’ further treatment was carried out by other doctors who did not know how the groups had been randomized. All further clinical data and outcomes were recorded by another attending physician who was unaware of how the groups had been randomized.

### 2.6. Intervention (tVNS)

For all study participants, tVNS was performed using the NovaTens device (Novameditek LLC, Moscow, Russia). Stimulation parameters were selected based on data from global scientific publications on the effectiveness and safety of tVNS. The stimulation frequency was 20 Hz, and the pulse duration was 200 μs. The intensity of the electric current for study subjects was selected individually, taking into account their pain perceptions: it was set one unit below the pain threshold.

Stimulation was performed in both study groups according to the same algorithm (Figure 1), specifically upon inclusion in the study (immediately upon admission) before the PCI. Then, tVNS continued to be applied throughout the entire PCI procedure and 30 min after its completion. After PCI, all patients were prescribed appropriate pharmaceutical treatments in compliance with the current clinical guidelines.

The PCI strategy, according to modern clinical guidelines, involved revascularization of the infarct-related artery. In cases of multivessel disease, after revascularization of the infarct-related artery, the heart team considered the suitability of subjecting the patient to CABG.

### 2.7. Sample Size Determination

As mentioned above, several similar studies have been conducted using animal models: MI was induced in animals, and the effects of tVNS were assessed. Only one randomized clinical trial was conducted on humans [17]. In this study, the authors assessed the in-hospital dynamics of concentrations of myocardial injury markers and inflammatory markers in 95 patients and observed statistically significant differences. Using the R. Lehr formula [18] and taking into account the possible loss of data over 12 months of monitoring (10% of the initial number of patients in the sample), the 80% power of this study, and the alpha α = 0.05, we calculated the required number of observations at the level of 105 patients. Hence, we planned to randomize 110 study participants.

### 2.8. Statistical Analysis

Statistical analyses were performed using STATISTICA^®^ 10.0 Statsoft (Tusla, OK, USA) and MedCalc version 23.0.1. (MedCalc Software Ltd., Ostend, Belgium) software. Quantitative data are presented as medians and interquartile ranges: Me (Q1; Q3). Categorical data are presented as counts and frequencies (n, %). To compare two independent samples, we employed the Mann–Whitney U test for quantitative variables and the Pearson’s chi-squared test for categorical variables. In addition, to compare the frequency of the studied events over time, we used the logrank test and constructed Kaplan–Meier survival curves. The difference between the groups was assumed to be statistically significant at *p* < 0.05.

### 2.9. Endpoints: Primary Outcome Measure

The primary endpoints were in-hospital mortality and 12-month (one-year) mortality.

### 2.10. Endpoints: Secondary Outcome Measure

The secondary endpoints were non-fatal cardiovascular events. The main in-hospital outcomes were pulmonary edema, cardiogenic shock, atrial fibrillation, ventricular tachycardia/fibrillation, accelerated idioventricular rhythm, second-degree and third-degree atrioventricular blocks, stroke or transient ischemic attack (TIA), temporary pacemaker implantation (TPI), and electric cardioversion. Furthermore, an evaluation and comparison of the levels of markers of myocardial damage (troponin, CPK-MB, and NT-proBNP) and inflammation (hs-CRP), heart rate, and episodes of cardiac rhythm disturbances at different times of hospitalization in the two groups were conducted (Figure 1).

Remote (one-year) non-fatal cardiovascular events included recurrent MI, stroke or TIA, and rehospitalization for heart failure. We also analyzed the composite endpoint, which combined major adverse cardiovascular events (MACEs): recurrent MI, stroke/TIA and overall mortality.

### 2.11. Monitoring

After discharge from the hospital, patients received treatment at their local outpatient clinics. They and their close relatives could engage in telephone contact with the study physician whenever questions arose. Data on rehospitalization and/or development of the studied events were collected using the city medical information system as well as through telephone communication between the study physician and the patient or his/her close relatives. No patients dropped out of observation during the 12-month period, and all information regarding all included patients was collected in full.

## 3. Results

### 3.1. Hospital Data

A total of 110 patients were randomized into the Active tVNS group (n = 55) and the Sham tVNS group (n = 55). One patient dropped out of the Active tVNS group because PCI was not performed. Thus, 109 patients were included in the final analysis (Figure 2).

The initial clinical data, along with laboratory, instrumental, and surgical data, did not differ statistically significantly between the groups (Table 1).

The laboratory and instrumental data for Day 1 are presented in Table 2. Troponin was observed to be at a statistically significantly lower level (albeit borderline) 6 h after hospitalization in the Active tVNS group compared to that in the Sham tVNS group (*p* = 0.048*). According to Holter monitoring, the daily heart rate and the number of paired ventricular ectopic (PVE) events were also lower in the Active tVNS group than in the Sham tVNS group (*p* < 0.001*, *p* = 0.002*).

After several days, the differences in the laboratory and instrumental parameters became more noticeable (Table 3). The levels of troponin, N-terminal prohormone of brain natriuretic peptide (NT-proBNP), and creatine phosphokinase-MB (CPK-MB) were statistically significantly lower in the Active tVNS group than in the Sham tVNS group (*p* < 0.001*, *p* = 0.013*, and *p* < 0.001*, respectively). The dynamics of their levels are shown in Figure 3, Figure 4 and Figure 5. The quantities of ventricular ectopic (VE) events and PVE events, including during Day 1 of ECG monitoring, were statistically significantly lower in the Active tVNS group than in the Sham tVNS group (*p* < 0.001* and *p* = 0.029*, respectively).

Hospital events are presented in Table 4. The incidence of in-hospital mortality, the development of cardiogenic shock, and third-degree atrioventricular block were statistically less frequent in the Active tVNS group than in the Sham tVNS group (*p* = 0.024*, *p* = 0.044*, and *p* = 0.013*, respectively).

Inpatient drug therapy was prescribed based on current clinical guidelines, which are presented in Appendix A.

### 3.2. Adverse Events During 12-Month Monitoring

In the first year after discharge from the hospital, seven patients died in both groups (three patients in the Active tVNS group and four patients in the Sham tVNS group), recurrent MI occurred in eight people (three and five, respectively), and three study subjects had strokes (one and two, respectively). When comparing the groups over the long term (separately for the studied outcomes) by constructing Kaplan–Meier survival curves, we found no statistically significant differences (Figure 6, Figure 7, Figure 8 and Figure 9). When comparing the groups by overall mortality taking into account in-hospital mortality, it was clear that there was a borderline value of statistical significance allowing the survival curves to diverge (Logrank test, *p* = 0.066) (Figure 6B). When comparing the groups by the composite endpoint taking into account hospital events, we confirmed the statistical significance of the differences (logrank test, *p* = 0.0016*) (Figure 10).

## 4. Discussion

In our opinion, we have obtained positive results. Already on Day 1, a statistically significant, albeit borderline, difference between the groups in terms of troponin levels was observed. Also, the Holter-monitoring data revealed that the daily heart rate and frequency of PVE events were lower in the Active tVNS group. That is, the early effects of tVNS were already noted on Day 1. A more pronounced difference between the groups was observed by Day 4: the levels of troponin, NT-proBNP, and CPK-MB were already even lower in the Active tVNS group. Differences in the frequency of ventricular arrhythmias remained, although the mean daytime and nighttime heart rates by Day 4 no longer exhibited statistically significant differences between the groups.

This was probably caused by the more frequent prescription of different medicamentous therapies in the Sham tVNS group. Amiodarone was prescribed more often in that group than in the Active tVNS group, wherein cardiac arrhythmias were less common (Appendix A).

In this study, we utilized fairly strict inclusion and exclusion criteria for selecting the admitted STEMI patients. This was due to the prognostic importance of the factors that have been shown to affect both the in-hospital and long-term outcomes we studied. Therefore, we only included patients with primary MI and primary PCI and excluded patients with a severe condition on admission (such as severe acute heart failure, severe arrhythmias, etc.). Obviously, this further affected our results in that the incidence of MACEs in the overall cohort was lower in the long term than what we would usually observe among patients in the first year after MI. This approach allowed us to form two homogeneous groups that were initially unquestionably similar in terms of the values of the clinical, laboratory, and instrumental parameters and exhibited differences solely in the investigated intervention (Active tVNS vs. Sham tVNS).

Since our study was conducted in a large city with a population above 1 million people where the emergency medical service is well organized, the median time from the onset of pain syndrome to hospital admission was about 1.5 h in both groups, which is a good result.

We would like to focus on the total duration of tVNS. It averaged just over an hour, and the upper quartile was 75 min. These data are important for future studies since the optimal threshold value of tVNS time for such patients had not been determined prior to our study. It is not yet known whether there is a direct relationship between improved treatment results and the duration of tVNS. We suggest that its duration should be increased in future studies, and the effect of such an increase needs to be investigated.

Only 5 patients out of the 109 included in this study died in the hospital. The causes of death were myocardial rupture for two patients, ventricular fibrillation for another two patients, and severe cardiogenic shock for one patient. All of them were in the Sham tVNS group. It was certainly surprising for us to obtain statistically significant differences in in-hospital mortality in this pilot study. Although the mechanisms of projective action have been described in sufficient detail in previous studies [19,20] and the favorable effect of tVNS has been proven in animal models [16], a clinical evaluation of VNS in MI is insufficient. We also revealed statistically significant differences in the incidence of cardiogenic shock: the incidence was lower in the Active tVNS group. Of course, mortality, cardiogenic shock, and pulmonary edema are directly related and linked to the process of dying. The incidence of pulmonary edema was also lower in the Active tVNS group, albeit without statistical significance. Another statistically significant difference between in-hospital outcomes was the incidence of the third-degree AV block, also seen more frequently in the Sham tVNS group.

As already mentioned in the Materials and Methods Section, optimal drug therapy was prescribed to all patients based on the accepted international guidelines. However, upon analyzing the prescription frequencies of different groups of drugs at the hospital, we discovered that there were significant differences in the frequencies of using beta-blockers and amiodarone. In the Active tVNS group, beta-blockers were prescribed more often and amiodarone was prescribed less often than in the Sham tVNS group. This is explained by the fact that arrhythmias (VE events, atrial fibrillation/atrial flutter) during hospitalization were more often observed in the Sham tVNS group (Table 3); consequently, doctors more often prescribed amiodarone as a pronounced antiarrhythmic and prescribed beta-blockers less often to these patients. Conversely, in the Active tVNS group, where arrhythmic events were less common, amiodarone was not so much in demand, while beta-blockers were prescribed more often.

In this study, we obtained important data on remote outcomes. To the best of our knowledge, this is the first study to evaluate the one-year outcomes of the clinical use of VNS in the acute period of MI. We observed boundary values of statistical significance with respect to the divergence of the curves of overall mortality and a statistically significant difference for the composite endpoint. When assessing the obtained curves, we noticed that the main contribution to the frequency of the studied events was certainly made by the in-hospital stage of treatment. This is especially true for mortality as an event. Indeed, from the standpoint of possible protective mechanisms, tVNS is more effective in the acute period of ischemia development because it suppresses oxidative stress processes and normalizes mitochondrial functions and fatty acid metabolism, among other mechanisms [21]. In other words, the earlier the vagus nerve activity is increased, the greater the benefit in acute ischemia. Therefore, the effect of tVNS is immediately obvious during the period in which a patient is being treated in the hospital. This may explain the discrepancy that was obtained from the long-term results: the frequency of MACEs, which were included in the combined endpoint, was more pronounced in the hospital period than in the long term. Therefore, the individual components of the combined endpoint were insignificant or of borderline insignificance in the long term, and the total frequency of MACEs in the long-term period was already statistically significantly different between the groups.

The pathophysiological concept of this study suggests that early tVNS through the efferent fibers of the vagus suppresses the active release of myocardial cytokines during the acute period of myocardial ischemia and during the restoration of blood flow after a PCI, minimizing the adverse effects of reperfusion injury (including by suppressing the active release of cytokines). Consequently, the suppression of the so-called “cytokine storm” by tVNS in the early period is manifested in a decrease in the area of ischemia and myocardial infarction, which certainly has a prognostically favorable effect on long-term events. In this regard, further studies with a larger sample size and a longer follow-up period are certainly needed.

The results of our study highlight the effect of VNS on the clinical outcomes of patients with acute STEMI. Reductions in the levels of myocardial injury biomarkers and a decrease in the incidence of ventricular arrhythmias resulting from reperfusion are important indicators of the effectiveness of this approach. tVNS may represent a beneficial strategy for reducing the size of the area affected by MI and diminishing the adverse effects of reperfusion, offering significant clinical implications. Besides that, the safety of tVNS makes it noninvasive and promising for use. Given the improvements in the long-term patient outcomes, such as improved survival and reduced recurrent cardiovascular events, it can be concluded that this method has noteworthy potential as an additional therapeutic tool in the complex treatment of STEMI.

### Limitations of This Study

The main limitation of this study is the small sample of patients. Although we calculated the number of subjects that needed to be included in this study, this calculation was based on data from only one clinical trial. Therefore, our findings definitely need to be confirmed, and a larger and more extensive multicenter study needs to be conducted.

Also, a limitation of this work is the lack of clear parameters for the most effective stimulation: frequency, pulse wavelength, intensity, and duration of stimulation. We applied the most commonly used stimulation parameters in the global literature. These data can also potentially influence the effectiveness of this method. In this regard, an internationally developed consensus protocol of guidelines for reporting tVNS studies should address these issues in the future [22].

## 5. Conclusions

An important result of this study was the statistically significant decrease in the primary endpoint, hospital mortality, in the group subjected to active vagal stimulation. The one-year results did not show significant statistically significant differences in the frequency of outcomes for individual events, but there was a significant difference in the combined endpoint.

The results of this study show the potential of using vagus nerve electrical stimulation in clinical practice, namely, as an additional therapeutic method within the framework of the complex treatment of STEMI. The confirmation of a significant reduction in the levels of myocardial damage biomarkers (troponin and creatine kinase MB) in the active vagus stimulation group, as well as the reduction in the frequency of ventricular arrhythmias, indicates the potential clinical advantages of this approach. The decrease in NT-proBNP levels in the first few days may be an indicator not only of a quantitative reduction in myocardial damage but also improved stability of cardiac function under neuromodulation exposure, suggesting a potential impact on patient survival and quality of life.

Therefore, further studies are needed in order to increase statistical power by increasing the number of patients. The relevance of further studies with expanded sampling, allowing more detailed verification of the findings, is critical. This is necessary to confirm the efficacy of tVNS and deeply analyze the safety and feasibility of its use in routine clinical practice.

## Figures and Tables

**Figure 1 jcm-14-01866-f001:**
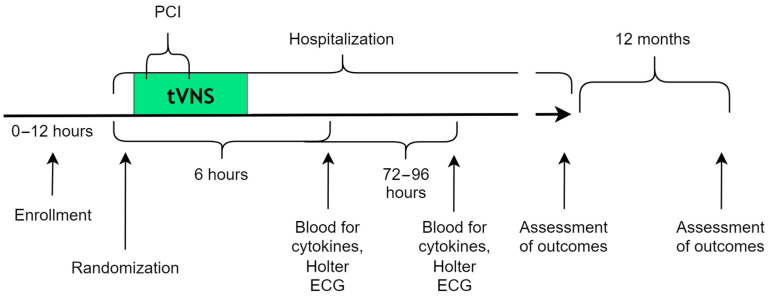
Diagram of the beginning and end of stimulation upon admission of a patient with MI.

**Figure 2 jcm-14-01866-f002:**
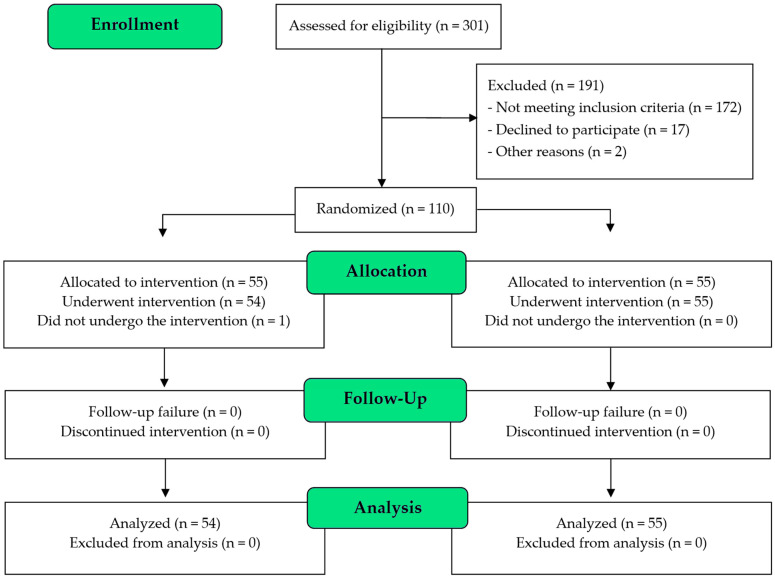
CONSORT diagram.

**Figure 3 jcm-14-01866-f003:**
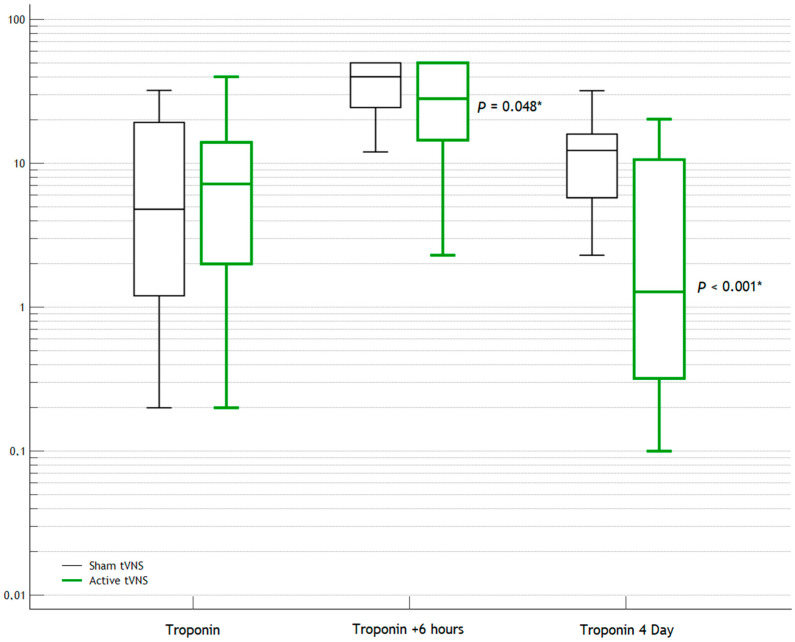
Dynamics of troponin levels. *—statistically significant differences.

**Figure 4 jcm-14-01866-f004:**
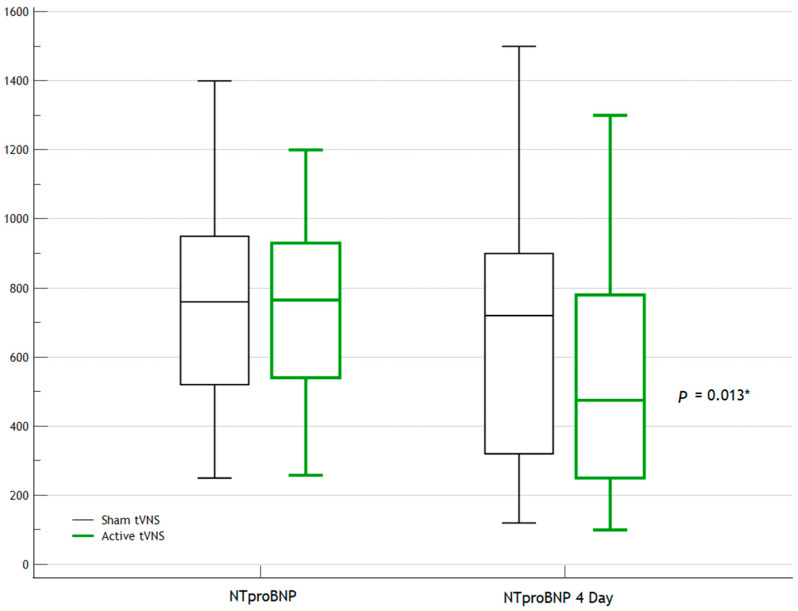
Dynamics of NT-proBNP levels. *—statistically significant differences.

**Figure 5 jcm-14-01866-f005:**
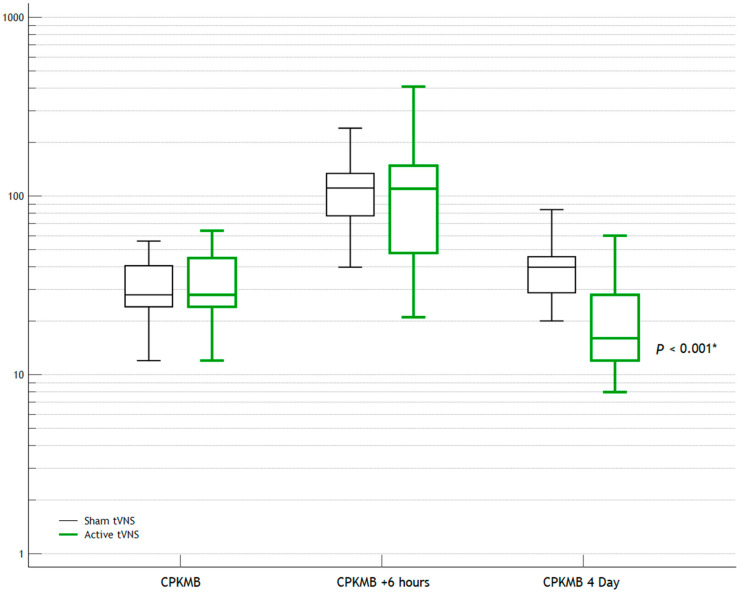
Dynamics of creatine phosphokinase-MB levels. *—statistically significant differences.

**Figure 6 jcm-14-01866-f006:**
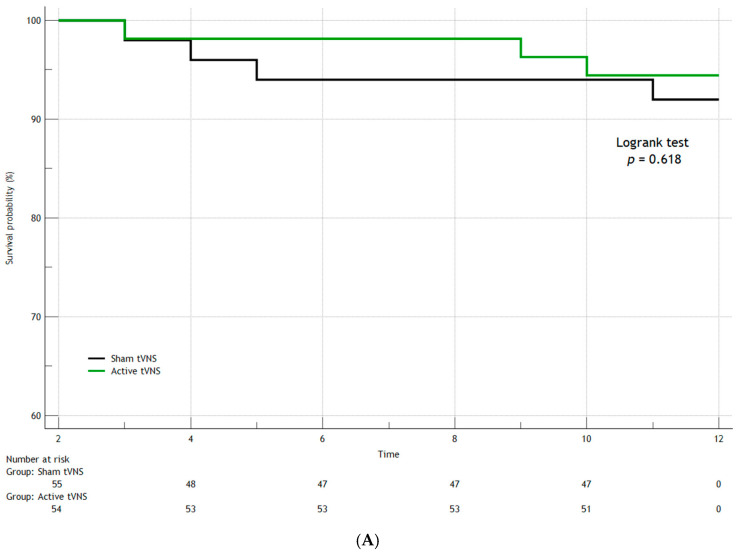
Kaplan–Meier survival curves for overall mortality (logrank test): (**A**) excluding in-hospital mortality; (**B**) including in-hospital mortality.

**Figure 7 jcm-14-01866-f007:**
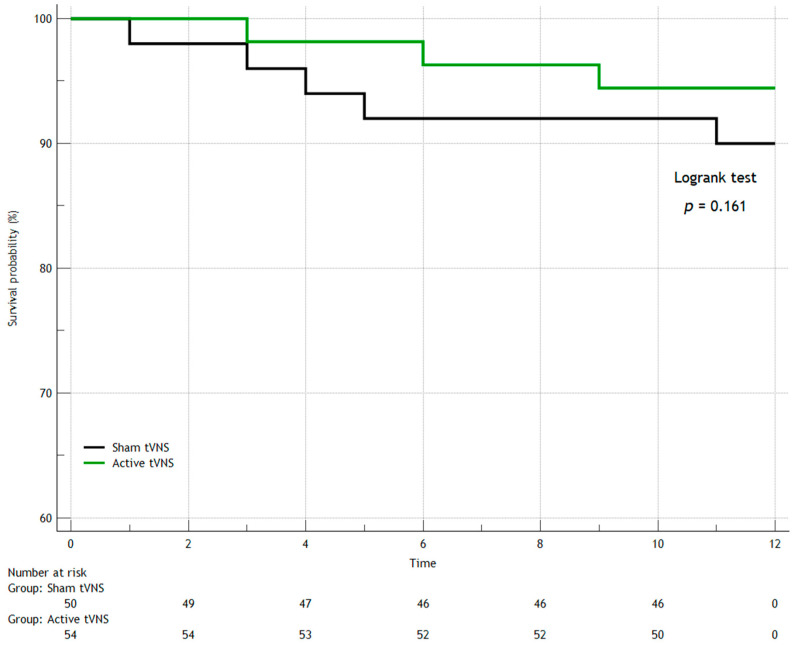
Kaplan–Meier survival curves for recurrent myocardial infarction.

**Figure 8 jcm-14-01866-f008:**
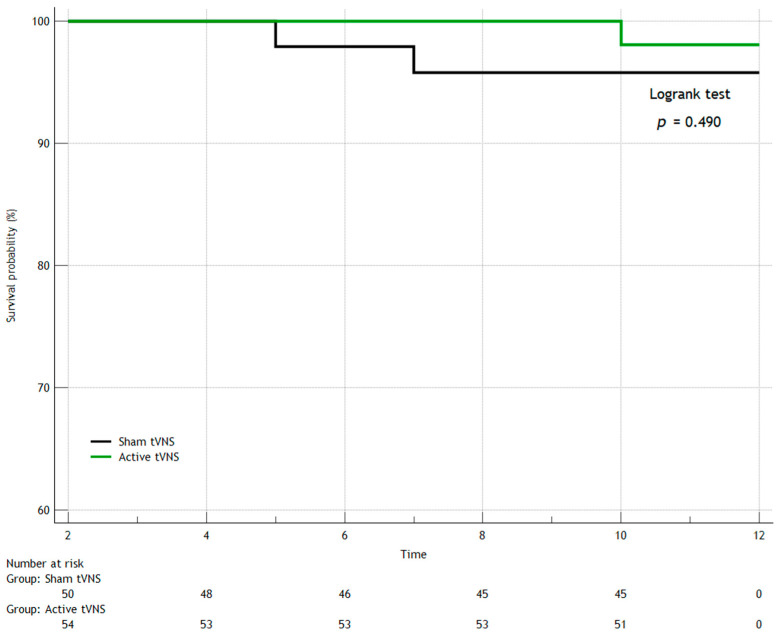
Kaplan–Meier survival curves for stroke/transient ischemic attack.

**Figure 9 jcm-14-01866-f009:**
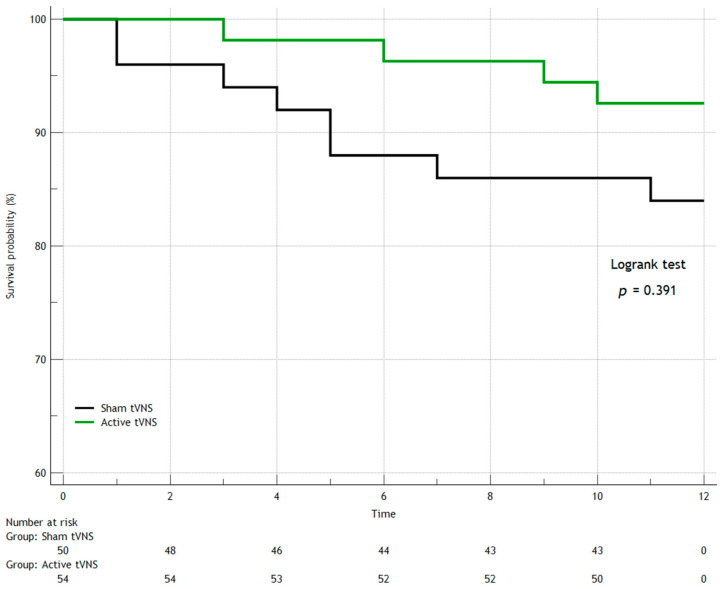
Kaplan–Meier survival curves for rehospitalization for heart failure.

**Figure 10 jcm-14-01866-f010:**
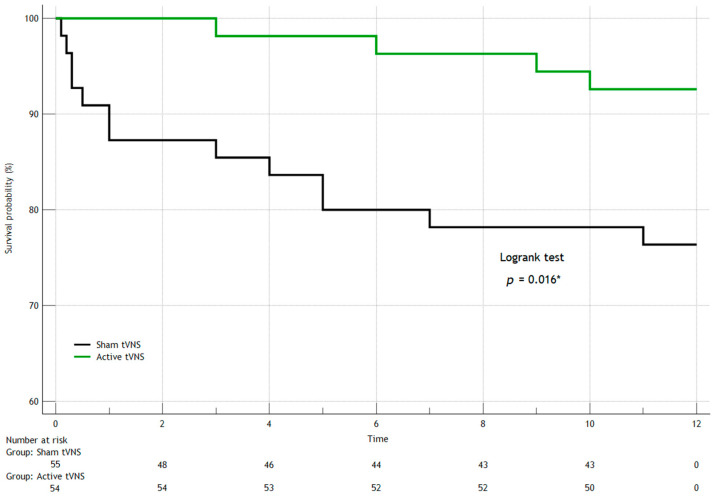
Kaplan–Meier survival curves for composite endpoint development taking into account hospital events (logrank test, *p* = 0.016 *). *—statistically significant differences.

**Table 1 jcm-14-01866-t001:** Basic parameters of patients, according to the initial data.

Parameters	Active tVNS (n = 54)	Sham tVNS (n = 55)	*p*
Clinical parameters of patients			
Male, n (%)	34 (63)	37 (67)	0.641
Age, yrs	67 (60; 72)	65 (55; 71)	0.117
Weight, kg	80 (70; 96)	85 (75; 90)	0.954
Height, cm	172 (166; 175)	173 (164; 178)	0.785
Angina pectoris, n (%)	15 (28)	15 (27)	0.956
Hypertension, n (%)	51 (94)	50 (91)	0.485
Diabetes, n (%)	9 (17)	18 (32)	0.054
Stroke, n (%)	1 (1.8)	2 (3.6)	0.578
PAD, n (%)	4 (7)	1 (1.8)	0.168
COPD, n (%)	0 (0)	0 (0)	-
Smoking, n (%)	11 (20)	12 (22)	0.857
Hospital admission data			
Pain-to-admission time, min	90 (60; 120)	80 (60; 120)	0.639
Door-to-balloon time, min	20 (15; 25)	20 (15; 25)	0.399
tVNS-to-balloon time, min	10 (8; 12)	10 (8; 15)	0.722
Total duration of tVNS, min	70 (65; 75)	65 (60; 75)	0.114
Troponin	6.2 (2; 14)	4.8 (1.2; 20)	0.813
CPK-MB	28 (24; 45)	28 (24; 41)	0.958
hs-CRP	22 (14; 22)	18 (12; 24)	0.072
NT-proBNP	765 (540; 930)	760 (520; 950)	0.978
ST2	60 (40; 124)	64 (28; 118)	0.750
WBC	7.2 (7.2; 10.2)	9.2 (6.3; 12.2)	0.624
HR	75 (65; 80)	80 (70; 85)	0.289
Operating room data			
LDA, n (%)	28 (52)	22 (40)	0.218
CA/OMA, n (%)	13 (24)	21 (38)	0.114
Right coronary artery, n (%)	13 (24)	15 (27)	0.706
Main left coronary artery, n (%)	1 (1.8)	1 (1.8)	1.000
Stenting of 2 arteries, n (%)	1 (1.8)	4 (7)	0.181

Note: tVNS, transcutaneous vagus nerve stimulation; PAD, peripheral artery disease; COPD, chronic obstructive pulmonary disease; CPK-MB, creatine phosphokinase-MB; hs-CRP, high-sensitivity C-reactive protein; NT-proBNP, N-terminal prohormone of brain natriuretic peptide; ST2, suppression of tumorigenicity 2 gene; WBCs, white blood cells; HR, heart rate; LDA, left descending artery; CA, circumflex artery; OMA, obtuse marginal artery.

**Table 2 jcm-14-01866-t002:** The laboratory and instrumental data from Day 1.

Parameters	Active tVNS (n = 54)	Sham tVNS (n = 55)	*p*
Troponin (+6 h)	28.2 (14.5; 50)	40 (24; 50)	0.048 *
CPK-MB (+6 h)	110 (48; 148)	111 (78; 134)	0.392
Hospital admission data			
LVEF, %	42 (35; 47)	45 (38; 48)	0.317
LA, mm	37 (35; 38)	38 (37; 39)	0.402
IVS, mm	10 (9; 10)	10 (9; 11)	0.269
Posterior wall, mm	9 (8; 11)	10 (9; 11)	0.089
LVEDD, mm	45 (42; 46)	45 (45; 52)	0.587
Holter			
VE events	19 (10; 23)	120 (2; 630)	0.110
PVE events	0 (0; 0)	0 (0; 40)	0.002 *
VT	0 (0; 0)	0 (0; 0)	-
VF, n (%)	0 (0)	1 (1.8)	0.330
Daytime HR	78 (74; 85)	86 (84; 92)	<0.001 *
Nighttime HR	68 (63; 72)	75 (70; 80)	<0.001 *

Note: CPK-MB, creatine phosphokinase-MB; LVEF, left ventricular ejection fraction; LA, left atrium; IVS, interventricular septum; LVEDD, left ventricular end-diastolic diameter; VE events, ventricular ectopic events; PVEs, paired ventricular ectopic events; VT, ventricular tachycardia; VF, ventricular fibrillation; HR, heart rate. *—statistically significant differences.

**Table 3 jcm-14-01866-t003:** The laboratory and instrumental data from Days 3–4.

Parameters	Active tVNS (n = 54)	Sham tVNS (n = 55)	*p*
Day 3			
hs-CRP	10 (10; 14)	15 (8; 20)	0.093
Day 4			
NT-proBNP	475 (250; 780)	720 (320; 900)	0.013 *
Troponin	1.3 (0.32; 10.2)	12.3 (5.6; 16)	<0.001 *
CPK-MB	16 (12; 28)	40 (29; 45)	<0.001 *
LVEF	46 (40; 50)	48 (42; 50)	0.300
VE events	0 (0; 0)	12 (0; 0)	<0.001 *
PVE events	0 (0; 0)	0 (0; 2)	0.029 *
VT	0 (0; 0)	0 (0; 0)	-
VF, n (%)	0 (0)	1 (1.8)	0.330
Daytime HR	72 (70; 76)	82 (58; 94)	0.179
Nighttime HR	60 (58; 65)	67 (50; 86)	0.596

Note: hs-CRP, high-sensitivity C-reactive protein; NT-proBNP, N-terminal prohormone of brain natriuretic peptide; CPK-MB, creatine phosphokinase-MB; LVEF, left ventricular ejection fraction; VE events, ventricular ectopic events; PVEs, paired ventricular ectopic events; VT, ventricular tachycardia; VF, ventricular fibrillation; HR, heart rate. *—statistically significant differences.

**Table 4 jcm-14-01866-t004:** Hospital outcomes.

Hospital Events	Active tVNS (n = 54)	Sham tVNS (n = 55)	*p*
In-hospital mortality, n (%)	0 (0)	5 (9)	0.024 *
Pulmonary edema, n (%)	2 (3.7)	6 (11)	0.153
Cardiogenic shock, n (%)	3 (5.6)	10 (18)	0.044 *
AF, n (%)	2 (3.7)	6 (11)	0.153
VT, n (%)	3 (5.6)	4 (7)	0.721
VF, n (%)	0 (0)	2 (3.6)	0.163
AIVR, n (%)	0 (0)	0 (0)	-
AV block 2, n (%)	1 (1.8)	4 (7)	0.181
AV block 3, n (%)	0 (0)	6 (11)	0.013 *
Stroke/TIA, n (%)	0 (0)	1 (1.8)	0.330
Pacemaker implantation, n (%)	0 (0)	3 (5.5)	0.085
Electric cardioversion, n (%)	0 (0)	2 (3.6)	0.163
In-hospital mortality, n (%)	0 (0)	5 (9)	0.024 *

Note: AF, atrial fibrillation; VT, ventricular tachycardia; VF, ventricular fibrillation; AIVR, accelerated idioventricular rhythm; AV block 2, second-degree atrioventricular block; AV block 3, third-degree atrioventricular block; TIA, transient ischemic attack. *—statistically significant differences.

## Data Availability

The parameters presented in this study are available on reasonable request from the corresponding author. The primary data are not publicly available due to the Scientific Research Institute of Regional Clinical Hospital #1’s policy regarding access to clinical data.

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
