# Peer review of "Low-Frequency Electrical Stimulation of the Auricular Branch of the Vagus Nerve in Patients with ST-Elevation Myocardial Infarction: A Randomized Clinical Trial"

_jcm, 2025, doi:10.3390/jcm14061866_

Round 1

Reviewer 1 Report

Comments and Suggestions for Authors

The authors present an interesting trial comparing the effects of transcutaneous vagus nerve stimulation (tVNS) in patients with ST elevation myocardial infarction (STEMI). They randomized 110 patients into tVNS arm and sham control group and looked at both in hospital and long term outcomes. I have some suggestions for the manuscript. 

1. In the abstract, describing the primary and secondary outcomes, would provide better clarity. A clearer mention of the specific outcomes under each category could enhance the readability and coherence of the abstract. 
2. Study Design: The study design is good and aims to study the effects of tVNS in a clinical setting. The use of an active transcutaneous vagus nerve stimulator in the intervention arm and the inclusion of a same control group are strengths of the design, providing good comparison. 
3. Sample Size Determination and Statistical Analysis: The sample size determination and statistical analysis is appropriate for the study design. Follow up data is good with long term follow up of one year. 
4. Results Section: The results are well-presented overall, but I recommend restructuring this section for improved clarity. It would be helpful to first describe the results in terms of primary and secondary outcomes before moving on to other findings. This would allow the reader to clearly understand the key objectives of the study and their associated outcomes.
5. The figures and tables are well-described and easy to interpret. 
6. I would advise avoiding phrases like "there is a tendency for the survival curves" in this context. The phrasing implies an incomplete conclusion and should be carefully reconsidered. It would be better to phrase this as a hypothesis, noting that the divergence of survival curves is suggestive but not definitive. Further investigation is required to confirm these findings.
7. Discussion Section: In the discussion, I recommend starting with a clear presentation of the main results, specifically highlighting the primary and secondary outcomes. Also recommend that the authors emphasize the potential clinical significance of the reduction in in-hospital mortality in the tVNS group. This finding is not sufficiently highlighted in the conclusions section. The authors should address this as a key result of the study, which could have important implications for clinical practice. 
Furthermore, the authors could elaborate on the potential mechanisms responsible for the observed reduction in mortality in the auricular stimulation group. This would add depth to the discussion and provide insights into the physiological effects of tVNS that might contribute to improved patient outcomes.
Can the authors link the findings regarding biomarkers to the results of the study? 
8. conclusion: The authors should incorporate the finding of reduced mortality more prominently as this is a significant finding. 
9. References: The references provided are adequate and relevant to the study.

Overall, the manuscript presents a well-executed trial with compelling results, especially in hospital mortality. The manuscript could be improved by emphasizing primary and secondary outcomes, discussing the in hospital mortality reduction 

Author Response

1. In the abstract, describing the primary and secondary outcomes, would provide better clarity. A clearer mention of the specific outcomes under each category could enhance the readability and coherence of the abstract.

We thank the reviewer for noting this. We have tried to correct this.

2. Study Design: The study design is good and aims to study the effects of tVNS in a clinical setting. The use of an active transcutaneous vagus nerve stimulator in the intervention arm and the inclusion of a same control group are strengths of the design, providing good comparison. 

We thank the reviewer for appreciating our study.

3. Sample Size Determination and Statistical Analysis: The sample size determination and statistical analysis is appropriate for the study design. Follow up data is good with long term follow up of one year. 

We thank the reviewer for noting this positively.

4. Results Section: The results are well-presented overall, but I recommend restructuring this section for improved clarity. It would be helpful to first describe the results in terms of primary and secondary outcomes before moving on to other findings. This would allow the reader to clearly understand the key objectives of the study and their associated outcomes.

On the one hand, we understand the logic of the reviewer. However, on the other hand, the primary point is mortality. And it doesn't seem optimal to start with it. In the article, we wanted to show the differences in their appearance in chronological order. Initially, the groups are comparable; after stimulation, the first differences are minimal on the first day, then they increase by 3-4 days. And then the mechanisms of hospital events are already clear and logical from the pathophysiology point of view. We would like to leave the description in chronological order, if the reviewer doesn't mind.

5. The figures and tables are well-described and easy to interpret. 

We thank the reviewer for noting this positively.

6. I would advise avoiding phrases like "there is a tendency for the survival curves" in this context. The phrasing implies an incomplete conclusion and should be carefully reconsidered. It would be better to phrase this as a hypothesis, noting that the divergence of survival curves is suggestive but not definitive. Further investigation is required to confirm these findings.

We thank the reviewer for noting this. We have tried to correct this.

7. Discussion Section: In the discussion, I recommend starting with a clear presentation of the main results, specifically highlighting the primary and secondary outcomes. Also recommend that the authors emphasize the potential clinical significance of the reduction in in-hospital mortality in the tVNS group. This finding is not sufficiently highlighted in the conclusions section. The authors should address this as a key result of the study, which could have important implications for clinical practice. 
Furthermore, the authors could elaborate on the potential mechanisms responsible for the observed reduction in mortality in the auricular stimulation group. This would add depth to the discussion and provide insights into the physiological effects of tVNS that might contribute to improved patient outcomes.
Can the authors link the findings regarding biomarkers to the results of the study? 

We thank the reviewer for noting this. We have tried to correct this.

8. conclusion: The authors should incorporate the finding of reduced mortality more prominently as this is a significant finding. 

We thank the reviewer for noting this. We have tried to correct this.

9. References: The references provided are adequate and relevant to the study.

We thank the reviewer for noting this positively.

Overall, the manuscript presents a well-executed trial with compelling results, especially in hospital mortality. The manuscript could be improved by emphasizing primary and secondary outcomes, discussing the in hospital mortality reduction.

We thank the reviewer for appreciating our study. We have highlighted all corrections in the article in green. 

Reviewer 2 Report

Comments and Suggestions for Authors

The manuscript entitled „Low-frequency electrical stimulation of the auricular branch of the vagus nerve in patients with ST-elevation myocardial infarction: А randomized clinical trial” was submitted to the Journal of Clinical Medicine for consideration.

The issue addressed in the manuscript in the era of successful interventional management of STEMI is of decreasing clinical importance, and the authors base the rationale for performing their analysis on the limited data about vagus nerve stimulation (VNS) studies observed in animal models of myocardial infarction. To date, a translation of VNS into human clinical practice has been limited, and the current study was aimed to fill this gap by evaluating the effect of transcutaneous VNS (tVNS) on outcomes in patients with ST-elevation myocardial infarction (STEMI).

In their manuscript, the authors conducted a blinded (single?), randomized, placebo-controlled trial with 110 STEMI patients. Participants were randomly assigned to either an active tVNS group (n=55) or a sham tVNS group (n=55), with the active group receiving stimulation of the tragus, and the sham group receiving stimulation only of the lobule. The results indicated that on the first day post-intervention, the active tVNS group showed lower troponin levels and fewer peri-procedural ventricular arrhythmias (PVE) compared to the sham group. The authors suggest that tVNS could potentially provide benefit in reducing myocardial injury and complications in STEMI patients.

The concept of the study, and the suggested influence of tVNS on the outcomes is based on the relatively short procedure (lasting max 30 minutes after PCI), affecting even long-term outcomes, without taking into consideration many elements potentially affecting survival. The HR is provided solely for day-time/night-time in Day 1, while there is no information about the acute effect of tVNS on the HR (no information on what was HR at the end of stimulation). 

The calculation of study size does not provide any statistical analyses, and rationale for its calculation - what would be the 80% power based on, and what was the estimated % of benefit for the VNS? In that context, a sample size of 110 patients may not be large enough to detect small but clinically significant differences, especially once >170 patients were not included due to not meeting inclusion criteria. There is no multivariable analysis to confirm the prognostic role of tVNS in STEMI. 

The angiographic characteristics look rather vague - there is no information what the „left coronary artery” means in the context of the LDA/Cx/OM. 

There are no definitions of Holter events among Methods, and no information on drugs potentially influencing heart rate (what were the beta-blocker doses utilized in the study?). There is no information on the strategy (upfront multivessel PCI in patients with multivessel CAD, staging of the procedure etc). 

The significant differences seem inconsistent - in one case, a clinically debatable difference of 0 vs 0 (PVE events) at day 1 is significant, while 19 vs 120 VEs are not significant, becoming significant on day 3 with 0 vs 12 VEs. Moreover, the clinical significance of such differences seems negligible. 

There is no information on the factors most importantly affecting long-term outcomes, including doses of statin therapy, implantation of ICD in patients with LVEF<35%, cardiac rehabilitation, or optimal treatment in patients who developed HF. 

Very rarely, the discussion can be started with „remarkable”, even in the most ground-breaking science the discussions are initiated in the following way. The supplementary data are written in Russian. 

Minor comments:
-there is no phrase as „severe CABG in 1 patient”

Author Response

R2
The manuscript entitled „Low-frequency electrical stimulation of the auricular branch of the vagus nerve in patients with ST-elevation myocardial infarction: А randomized clinical trial” was submitted to the Journal of Clinical Medicine for consideration.

Response:

We thank the reviewer for appreciating our study. We have highlighted all corrections in the article in green.

Comments 1:
The issue addressed in the manuscript in the era of successful interventional management of STEMI is of decreasing clinical importance, and the authors base the rationale for performing their analysis on the limited data about vagus nerve stimulation (VNS) studies observed in animal models of myocardial infarction. To date, a translation of VNS into human clinical practice has been limited, and the current study was aimed to fill this gap by evaluating the effect of transcutaneous VNS (tVNS) on outcomes in patients with ST-elevation myocardial infarction (STEMI).

Response 1: The emergence of geographically accessible regional vascular centers for interventional STEMI treatment has reduced hospital mortality and improved long-term survival of such patients. However, despite this, the STEMI problem has not been fully resolved. Technical, logistic, and clinical limitations still remain relevant. Therefore, the study of new approaches and treatment methods is always important and should be considered.

Comments 2:
In their manuscript, the authors conducted a blinded (single?), randomized, placebo-controlled trial with 110 STEMI patients. Participants were randomly assigned to either an active tVNS group (n=55) or a sham tVNS group (n=55), with the active group receiving stimulation of the tragus, and the sham group receiving stimulation only of the lobule. The results indicated that on the first day post-intervention, the active tVNS group showed lower troponin levels and fewer peri-procedural ventricular arrhythmias (PVE) compared to the sham group. The authors suggest that tVNS could potentially provide benefit in reducing myocardial injury and complications in STEMI patients.

Response 2:

This was a single-blind study. It was difficult to ensure double-blindness due to technical limitations. Patients were urgently admitted to the hospital at any time of the day.
The on-call doctor in the emergency cardiology department, who participated in the study, decided whether the patient met all the inclusion and exclusion criteria. Therefore, after randomization, the same doctor installed the tVNS device and accompanied the patient throughout the stimulation procedure. Of course, he knew which intervention group the patient belonged to. However, this did not affect the assessment of the results, as the patient's further treatment was carried out by other doctors who did not know the randomization group. All further clinical data and outcomes were recorded by another attending physician who was unaware of the randomization group.

Comments 3:
The concept of the study, and the suggested influence of tVNS on the outcomes is based on the relatively short procedure (lasting max 30 minutes after PCI), affecting even long-term outcomes, without taking into consideration many elements potentially affecting survival. The HR is provided solely for day-time/night-time in Day 1, while there is no information about the acute effect of tVNS on the HR (no information on what was HR at the end of stimulation).

Response 3: The pathophysiological concept of the study suggests that early tVNS through vagus efferent fibers suppresses the active release of myocardial cytokines in the acute period of myocardial ischemia and during blood flow recovery after PCI, minimizing the adverse effects of reperfusion damage (also through suppression of active cytokine release). Consequently, suppression of the so-called "cytokine storm" through tVNS in the early period manifests as a reduction in the ischemia and myocardial infarction zone, which undoubtedly has a prognostically favorable effect on long-term outcomes.
As for other factors that could potentially influence survival, they were assessed and did not differ between the groups. The design of the randomized study suggests that confounding factors are blocked. Initially, patients were comparable in all basic clinical, instrumental, and laboratory parameters. Medication therapy was also prescribed according to current clinical guidelines in both groups. We presented the results of heart rate, which were assessed and based on daily Holter monitoring data, as they reflect weighted and average values for the day: during the day and at night. Analyzing heart rate immediately after the procedure, in our opinion, does not make any sense, as minute heart rate measurements are susceptible to significant fluctuations and may be unreliable.

Comments 4:
The calculation of study size does not provide any statistical analyses, and rationale for its calculation - what would be the 80% power based on, and what was the estimated % of benefit for the VNS? In that context, a sample size of 110 patients may not be large enough to detect small but clinically significant differences, especially once >170 patients were not included due to not meeting inclusion criteria. There is no multivariable analysis to confirm the prognostic role of tVNS in STEMI.

Response 4: The sample size calculation, as we described in the article, is based on the work of Lilei Yu (reference 17). In this study, the authors evaluated the hospital dynamics of myocardial injury and inflammatory markers in 95 patients and found statistically significant differences. There are currently no clinical studies that have assessed long-term mortality and other long-term events. Therefore, these frequencies simply cannot be obtained to estimate the effect dispersion necessary for the formula. As for the comment about why we used the 80% power, it is based on the work of R. Lehr, where he justifies the choice of these parameters. Robert Lehr, Sixteen S-squared over D-squared: A relation for crude sample size estimates. https://doi.org/10.1002/sim.4780110811 There are various calculators available for determining the necessary sample size. For example, this one - https://www.sealedenvelope.com/power/binary-superiority/ - has a default Power (1-beta) value of 90%. The debate about which calculator is the most optimal continues to this day. We often use the R. Lehr formula, as it is a familiar tool to us. Regarding the 170 patients who were not included in the study, our comment is as follows: We selected fairly stringent inclusion and exclusion criteria. We included patients only with primary MI and primary PCI and excluded patients with severe conditions upon admission (severe acute heart failure, severe cardiac arrhythmias, etc.). This approach allowed us to form, firstly, two homogeneous groups that were initially absolutely comparable in clinical, laboratory, and instrumental data and, secondly, these groups did not have a severe condition, which allowed us to focus on studying the effect of tVNS. If we included severe patients, the severity of their condition could overshadow the effect of tVNS stimulation. A multifactorial analysis in this case does not make sense to conduct, as we have a randomized, not a cohort study. We didn’t look for predictors of the development of the outcomes. We studied the effects of tVNS in randomized patient groups, assuming that confounding factors were blocked by randomization.

Comments 5:
The angiographic characteristics look rather vague - there is no information what the „left coronary artery” means in the context of the LDA/Cx/OM.

Response 5:
This is indeed an inaccurate translation. We’ve corrected it to “Main left coronary artery”.

Comments 6:
There are no definitions of Holter events among Methods, and no information on drugs potentially influencing heart rate (what were the beta-blocker doses utilized in the study?). There is no information on the strategy (upfront multivessel PCI in patients with multivessel CAD, staging of the procedure etc).

Response 6: We have corrected this section. We provided an analysis of Holter data in the "Material and Methods" section. All medication therapy is briefly described in the article and shown in additional materials in Table S1. It includes groups of drugs without doses. Patterns from types of beta-blockers and doses, as well as from other drugs, were not included in the goals of the work. These are quite heterogeneous data. Doses were selected individually according to clinical parameters, in accordance with clinical guidelines.
The intervention strategy, according to modern clinical guidelines, involved revascularization of the infarct-related artery. In case of multivessel disease, after revascularization of the infarct-related artery, the patient was considered by the heart team for CABG. In our study, there were no patients with multivessel disease included, which is likely due to the inclusion and exclusion criteria. We had only 1 patient whom 2 stents were implanted at once in the circumflex artery and Right coronary artery. The 1 stent was implanted to the remaining patients. We have added this information to the article.

Comments 7:
The significant differences seem inconsistent - in one case, a clinically debatable difference of 0 vs 0 (PVE events) at day 1 is significant, while 19 vs 120 VEs are not significant, becoming significant on day 3 with 0 vs 12 VEs. Moreover, the clinical significance of such differences seems negligible.

Response 7:
This is because the data were extremely non-normally and asymmetrically distributed. Therefore, non-parametric statistics were used - the Mann–Whitney U test comparison method. If you pay attention, in the Sham tVNS group, the upper quartile of PVE frequency was 40, while in the Active tVNS group it was -0. There was still some dispersion, and the comparison was significant. With VE frequency, the dispersion was less pronounced, so there were no significant differences. Of course, if we take into account only the median values, the data seem strange. It is important to consider the data scatter in this case. Fortunately, there are special calculation programs, and we do not calculate manually. For visual representation, we would like to show a screenshot from the STATISTICA® 10.0 Statsoft program.
The figures (Attached) show the data scatter and the result of comparing the two groups by VE and PVE parameters. 

Comments 8:
There is no information on the factors most importantly affecting long-term outcomes, including doses of statin therapy, implantation of ICD in patients with LVEF<35%, cardiac rehabilitation, or optimal treatment in patients who developed HF.

Response 8: Indeed, we did not track such detailed information (daily doses of statins, schemes and methods of cardiac rehabilitation) in the long-term period. It is very complicated and was not part of the objectives of our study. Patients after discharge from the hospital received medication therapy according to modern clinical guidelines. It is very difficult to track the doses they were prescribed on an outpatient basis. The patients were initially similar in groups according to clinical and instrumental data, and the severity of their condition.

The study was originally designed to be randomized, meaning that selection bias between treatment groups is minimized. This is intended to ensure that all factors affecting long-term outcomes are equally distributed among the groups.

Comments 9:
Very rarely, the discussion can be started with „remarkable”, even in the most ground-breaking science the discussions are initiated in the following way. The supplementary data are written in Russian.

Response 9: Although different countries have various forms of expressing their opinion, especially since this is the "Discussion" section, we considered the Reviewer's opinion and replaced this word.

Comments 10:
Minor comments: -there is no phrase as „severe CABG in 1 patient”

Response 10: Thank you for bringing this to our attention. This was a translator's mistake. We meant cardiogenic shock, not CABG. We've corrected it.

Reviewer 3 Report

Comments and Suggestions for Authors

The manuscript presents a well-structured randomized controlled trial (RCT) investigating the effect of transcutaneous vagus nerve stimulation (tVNS) in patients with ST-elevation myocardial infarction (STEMI). The study is well-conceived, and the results suggest potential clinical benefits of tVNS in reducing myocardial damage and improving short-term outcomes. 

The study calculates a required sample size of 105 based on a previous trial. However, the rationale for this number should be elaborated further, especially considering the relatively small sample size and potential type I and II errors.

The manuscript states that "statistical significance was revealed by Composite endpoint (Logrank test, p = 0.0016*)," but it remains unclear whether adjustments for multiple comparisons were performed.

The study notes that Kaplan-Meier survival curves did not show significant differences in long-term mortality (p=0.066). The authors suggest that tVNS benefits may be limited to the acute phase. This should be more explicitly discussed, especially in terms of potential mechanistic explanations.

The composite endpoint is significant, but individual components are not. This discrepancy warrants discussion regarding its clinical relevance.

The study is described as "blinded," but more details on the blinding process should be provided. Were the clinicians assessing outcomes blinded to the intervention groups?

Sham stimulation is performed at the earlobe, which lacks vagal innervation. However, could there be an unintentional bias in patient perception or physician handling?

The study reports that beta-blockers and amiodarone were prescribed at different rates between the groups. While the authors attribute this to different arrhythmic burdens, this introduces a confounder. Could a propensity-matched analysis or sensitivity analysis be conducted to assess the impact of these differences?

Were there differences in antithrombotic therapy or other medications that could have influenced the outcomes?

The proposed mechanism of action for tVNS is through anti-inflammatory and cardioprotective pathways. However, inflammatory biomarkers (e.g., hs-CRP) did not show significant differences. More discussion is needed regarding whether tVNS exerts its primary effect through autonomic modulation rather than direct anti-inflammatory effects.

The manuscript states that "no patients dropped out of observation during 12 months." However, adverse event reporting and potential side effects of tVNS should be elaborated upon.

Author Response

The manuscript presents a well-structured randomized controlled trial (RCT) investigating the effect of transcutaneous vagus nerve stimulation (tVNS) in patients with ST-elevation myocardial infarction (STEMI). The study is well-conceived, and the results suggest potential clinical benefits of tVNS in reducing myocardial damage and improving short-term outcomes. 

- We thank the reviewer for appreciating our work.

1. The study calculates a required sample size of 105 based on a previous trial. However, the rationale for this number should be elaborated further, especially considering the relatively small sample size and potential type I and II errors.

- The sample size calculation, as we described in the article, is based on the work of Lilei Yu (reference 17). In this study, the authors evaluated the hospital dynamics of myocardial injury and inflammatory markers in 95 patients and found statistically significant differences. There are currently no clinical studies that have assessed long-term mortality and other long-term events. Therefore, these frequencies simply cannot be obtained to estimate the effect dispersion necessary for the formula.

We used the 80% power. It is based on the work of R. Lehr, where he justifies the choice of these parameters. Robert Lehr, Sixteen S-squared over D-squared: A relation for crude sample size estimates. https://doi.org/10.1002/sim.4780110811
There are various calculators available for determining the necessary sample size. For example, this one - https://www.sealedenvelope.com/power/binary-superiority/ - has a default Power (1-beta) value of 90%. The debate about which calculator is the most optimal continues to this day. We often use the R. Lehr formula, as it is a familiar tool to us.

2. The manuscript states that "statistical significance was revealed by Composite endpoint (Logrank test, p = 0.0016*)," but it remains unclear whether adjustments for multiple comparisons were performed.

- We compared the frequency of the studied events over time using the Logrank test with the construction of Kaplan-Meyer survival curves for two groups (Active tVNS and Sham tVNS). The composite combined endpoint united major adverse cardiovascular events (MACE – recurrent myocardial infarction, stroke/TIA, and overall mortality). And we also looked at the difference in the "MACE events" over time using the Logrank test with the construction of Kaplan-Meyer survival curves for the two groups. In this situation, we do not have several groups, we did not use multiple comparisons. To be honest, we find it difficult to understand what the reviewer meant about the adjustments for multiple comparisons.

- A multifactorial analysis in this case does not make sense to conduct, as we have a randomized, not a cohort study. We didn’t look for predictors of the development of the outcomes. We studied the effects of tVNS in randomized patient groups, assuming that confounding factors were blocked by randomization.

3. The study notes that Kaplan-Meier survival curves did not show significant differences in long-term mortality (p=0.066). The authors suggest that tVNS benefits may be limited to the acute phase. This should be more explicitly discussed, especially in terms of potential mechanistic explanations.

- We have tried to supplement the manuscript in this direction.

4. The composite endpoint is significant, but individual components are not. This discrepancy warrants discussion regarding its clinical relevance.

- We've added this to the discussion.

5. The study is described as "blinded," but more details on the blinding process should be provided. Were the clinicians assessing outcomes blinded to the intervention groups?

- We have completed the manuscript in this direction.

6. Sham stimulation is performed at the earlobe, which lacks vagal innervation. However, could there be an unintentional bias in patient perception or physician handling?

- In fact, patients are not well aware of the method of vagus nerve stimulation and the anatomical features of the location of its fibers in the ear area. As for the awareness of the doctor, we have described this information in more detail in the second version of the article.

- This was a single-blind study. It was difficult to ensure double-blindness due to technical limitations. Patients were urgently admitted to the hospital at any time of the day. The on-call doctor in the emergency cardiology department, who participated in the study, decided whether the patient met all the inclusion and exclusion criteria. Therefore, after randomization, the same doctor installed the tVNS device and accompanied the patient throughout the stimulation procedure. Of course, he knew which intervention group the patient belonged to. However, this did not affect the assessment of the results, as the patient's further treatment was carried out by other doctors who did not know the randomization group. All further clinical data and outcomes were recorded by another attending physician who was unaware of the randomization group.

7. The study reports that beta-blockers and amiodarone were prescribed at different rates between the groups. While the authors attribute this to different arrhythmic burdens, this introduces a confounder. Could a propensity-matched analysis or sensitivity analysis be conducted to assess the impact of these differences?

- We understand the reviewer's question. That's why we have such a comment about it. The study was initially randomized. Using PSM in a randomized trial is pointless.  The patients were initially similar in groups according to clinical and instrumental data, and the severity of their condition. The study was originally designed to be randomized, meaning that selection bias between treatment groups is minimized. This is intended to ensure that all factors affecting long-term outcomes are equally distributed among the groups.

8. Were there differences in antithrombotic therapy or other medications that could have influenced the outcomes?

- All medication therapy is briefly described in the article and shown in supplementary materials in Table S1.

9. The proposed mechanism of action for tVNS is through anti-inflammatory and cardioprotective pathways. However, inflammatory biomarkers (e.g., hs-CRP) did not show significant differences. More discussion is needed regarding whether tVNS exerts its primary effect through autonomic modulation rather than direct anti-inflammatory effects.

Indeed, hs-CRP did not differ in the groups after stimulation. We attribute this to the short-term nature of the stimulation. However, the studies in which tVNS showed its anti-inflammatory effect had a different design with a long period of daily stimulation (for example, in CHF or rheumatoid arthritis).

For example: Marsal, S. et al. (2021). Non-invasive vagus nerve stimulation for rheumatoid arthritis: a proof-of-concept study. The Lancet. Rheumatology3(4), e262–e269. https://doi.org/10.1016/S2665-9913(20)30425-2 

Verrier, R. et al. (2022). Multifactorial Benefits of Chronic Vagus Nerve Stimulation on Autonomic Function and Cardiac Electrical Stability in Heart Failure Patients With Reduced Ejection Fraction. Frontiers in physiology13, 855756. https://doi.org/10.3389/fphys.2022.855756

- We've added this to the discussion.

10. The manuscript states that "no patients dropped out of observation during 12 months." However, adverse event reporting and potential side effects of tVNS should be elaborated upon.

-It's not entirely clear what the reviewer meant. The stimulation was only 60-70 minutes on average (Table 1, Figure 1). It wasn't long-lasting. Patients did not use vagal stimulation for 12 months after discharge. There could not be any side effects in the distant preiod.
- Again, if we take other tVNS studies with a long period of daily stimulation. These papers mention side effects in the form of local skin irritation with prolonged daily use. But this does not apply to our research design.

We thank the reviewer for appreciating our study. We have highlighted all corrections in the article in green. 

Round 2

Reviewer 3 Report

Comments and Suggestions for Authors

The authors answered all my comments.